# Large-scale mass wasting in the western Indian Ocean constrains onset of East African rifting

Vittorio Maselli [1✉], David Iacopini[2,3], Cynthia J. Ebinger[4], Sugandha Tewari[5], Henk de Haas [6], Bridget S. Wade [7], Paul N. Pearson [8], Malcom Francis[5], Arjan van Vliet[9], Bill Richards[1] & Dick Kroon[10]

Faulting and earthquakes occur extensively along the flanks of the East African Rift System, including an offshore branch in the western Indian Ocean, resulting in remobilization of sediment in the form of landslides. To date, constraints on the occurrence of submarine landslides at margin scale are lacking, leaving unanswered a link between rifting and slope instability. Here, we show the first overview of landslide deposits in the post-Eocene stratigraphy of the Tanzania margin and we present the discovery of one of the biggest landslides on Earth: the Mafia mega-slide. The emplacement of multiple landslides, including the Mafia mega-slide, during the early-mid Miocene is coeval with cratonic rifting in Tanzania, indicating that plateau uplift and rifting in East Africa triggered large and potentially tsunamigenic landslides likely through earthquake activity and enhanced sediment supply. This study is a first step to evaluate the risk associated with submarine landslides in the region.

[1] Department of Earth and Environmental Sciences, Life Sciences Centre, Dalhousie University, Halifax, NS B3H 4R2, Canada. [2] Dipartimento di Scienze della Terra, dell'Ambiente e delle Risorse, Università degli Studi di Napoli Federico II, Naples 80126, Italy. [3] School of Geosciences, University of Aberdeen, Aberdeen AB24 3FX, United Kingdom. [4] Department of Earth and Environmental Sciences, Tulane University, New Orleans, LA 70118, United States of America. [5] WesternGeco, Gatwick RH6 0NZ, United Kingdom. [6] National Marine Facilities, Royal Netherlands Institute for Sea Research and Utrecht University, Utrecht, The Netherlands. [7] Department of Earth Sciences, University College London, London WC1E 6BT, United Kingdom. [8] School of Earth and Ocean Sciences, Cardiff University, Cardiff CF10 3AT, United Kingdom. [9] Royal Dutch Shell, 2596 HR, The Hague, The Netherlands. [10] School of GeoSciences, University of Edinburgh, Edinburgh EH9 3FE, United Kingdom. ✉email: vittorio.maselli@dal.ca

Landslides and soil instability phenomena are widespread along the broad uplifted areas of the seismically and volcanically active East African Rift System (EARS), representing a serious hazard for many people from Eritrea to Mozambique[1]. Landslides also occur in the submarine environment[2], where they are readily imaged by marine seismic reflection data[3]. Submarine landslides represent a serious threat to engineered sea-floor structures and to coastal societies due to their potential in generating catastrophic tsunami waves[4]. In the western Indian Ocean, the spatial distribution and the timing of emplacement of submarine landslides are poorly understood, particularly with respect to uplift and tectonism of the EARS. This knowledge gap negatively affects the development of geohazard assessments for the region. Different mechanisms have been proposed as potential triggers and preconditioning factors of slope instability, including distant and local earthquakes, high rates of sediment supply, overpressure, and over-steepening of the sea floor due to tectonic movements[5–8]. More than 3400 earthquakes of magnitude $4.0 \le Mw \le 7.0$ have been recorded instrumentally since 1976 in East and Central Africa, and along an offshore branch of the EARS in the western Indian Ocean[9–11] (Fig. 1). Border faults within the EARS are unusually long (50–120 km), and they may rupture for the entire length generating M > 7 earthquakes, as occurred in 1910[9,12]. The passage of seismic waves from regional earthquakes may also dynamically trigger earthquakes along steep shelf margins[5]. To date, the relationship between EARS seismicity and the instability of the continental margin of East Africa is unknown, on both long-term and short-term time scales. Here we present the first overview of submarine landslide deposits in the post-Eocene stratigraphy of the continental margin of Tanzania in the western Indian Ocean, and we link their timing to the onset of uplift and rifting in East and Central Africa. In so doing, we report the discovery of a giant submarine landslide deposit that we name the Mafia mega-slide after the island located in its proximity. We use recently acquired geophysical and well data to test the hypothesis that the Mafia mega-slide was triggered by one or more, regional or local, earthquakes and enhanced sediment supply to the western Indian Ocean during the early phase of plateau uplift and rifting in East Africa. We discuss the results obtained to understand the link between cratonic rifting and slope instability, and to quantify the impact of this large-scale mass wasting event on the depositional processes shaping the continental slope of the western Indian Ocean.

## Results

**Faulting and magmatism in the East African Rift System**. At present, active faulting and magmatism occur across a large part of the African continent between the Horn of Africa southwest to the Okavango region of Botswana, east to Madagascar, and southeast to South Africa. Growing datasets indicate the diachroneity of magmatism, plateau uplift, and basin and flank formation spanning at least 40 million years to present day. Offshore data provide key new insights into the timing of rift initiation[13–15].

Magmatism in the EARS initiated in southwest Ethiopia and northern Kenya triggered by an initial mantle plume head impinging at the base of the African lithosphere at ~40 Ma[16–18]. Provenance analyses of Nile delta sequences indicate that the Ethiopian plateau uplift had occurred by about 30 Ma[19]. By ~31 Ma, extension in the Gulf of Aden and southern Red Sea was accompanied by widespread flood magmatism[20], whereas rifting south of the Afar depression initiated diachronously between 25 Ma and present day[21,22]. Rift-related volcanism started at 25–20 Ma in sectors of the eastern, western, and main Ethiopian arms of the EARS[21,23,24], but the timing of basin formation is hampered by the lack of well data. Some of the oldest ages come from the Rukwa–Malawi Rift region in the western branch of the EARS[25], at the headwaters of the Ruaha and Rufiji River drainages. A rift lake system had initiated by ~24.9 Ma when volcanic ash covered parts of the Rukwa–Malawi Rift region[26], and magmatism in the Rungwe Volcanic Province commenced by ~19 Ma[24,27]. Although much of the present-day Western Rift basin and flank morphology and magmatism developed in the past 15 million years[28–30], some extension occurred during the initial magmatic phase[24].

Plateau uplift south of Ethiopia diverted drainage eastward to the Indian Ocean and westward to the Congo basin, but the timing of plateau uplift remains debated, in large part owing to challenges in separating rift-flank uplift from plateau uplift[31–33]. It has been estimated through a comparison of observed and modelled carbonate compensation depths that there is about 650 m of dynamic uplift of the East African continental margin associated with the plume province starting at 25 Ma[15]. Lava flow geometry documents the existence of broad uplift near the Kenya-Tanzania border by 13.5 Ma[34]. Uplift of central Africa west of Lake Malawi occurred by 5 Ma when surface faulting initiated[35].

Eastward flowing rivers drain Permo–Triassic, Paleogene, and Neogene sedimentary basins between Lakes Malawi (Nyasa) and Rukwa[36]. Two major river systems drain the central and southern Tanzania: the Ruaha–Rufiji and the Rovuma Rivers (Fig. 1). The Ruaha River headwaters lie in the uplifted flanks of the Permo–Triassic and Paleogene–Recent Rukwa and Usangu Rifts, and the southern edge of the uplifted Tanzania craton, whereas the Rufiji River headwaters lie in the uplifted flanks of the tectonically active Malawi and Kilombero Rifts, and the ~19 Ma–Recent Rungwe Volcanic Province (Fig. 1). The Ruaha River transects the diffuse northern boundary of the Rovuma microplate (Fig. 1), interpreted from seismicity and sparse geodetic data[33,37,38], before its confluence with the Rufiji River. The Rufiji River delta forms landward of Mafia Island, a topographic high related to an inversion structure[39], which is part of the modern continental shelf (Fig. 1). The Rovuma River originates from the uplifted flanks of the central Malawi Rift, flowing along Permo–Triassic basins. The Rovuma River fed the paleo-Rovuma delta since the late Oligocene—early Miocene[40,41].

**Stratigraphy of the Tanzania margin**. The present study focuses on the post-Eocene stratigraphy, up to the modern sea floor, of the western Indian Ocean offshore Tanzania, which contains deposits of the Ruaha–Rufiji and Rovuma Rivers (Fig. 1). Overall, the post-Eocene succession comprises a series of slope channel complexes, vertically stacked, or characterized by high angles of southerly progradation that reflect significant sediment supply from the Ruaha–Rufiji and Rovuma Rivers and the influence of northward directed bottom currents[42,43] (Supplementary Note 1 and Supplementary Fig. 1). Moving upward in the stratigraphy, the development of channel complexes became affected by the topography generated by Neogene to Quaternary tectonic activity in the offshore branch of EARS[43], which led to the formation of the Kerimbas Graben and uplift of the Davie Ridge[13,14].

**Chronology of key stratigraphic markers**. In order to reconstruct the different phases of margin evolution, we identified and correlated regionally four seismic horizons, which are tied to two exploration wells (Figs. 2–6). The deepest seismic horizon, M1, corresponds to a high-amplitude reflection couplet that ties with Well-1 and Well-2 at a true vertical depth below mean sea level (ssTVD) of 2272 and 3505 m, respectively (Fig. 2). Horizon M2 is marked by a negative reflection, laterally changing in amplitude,

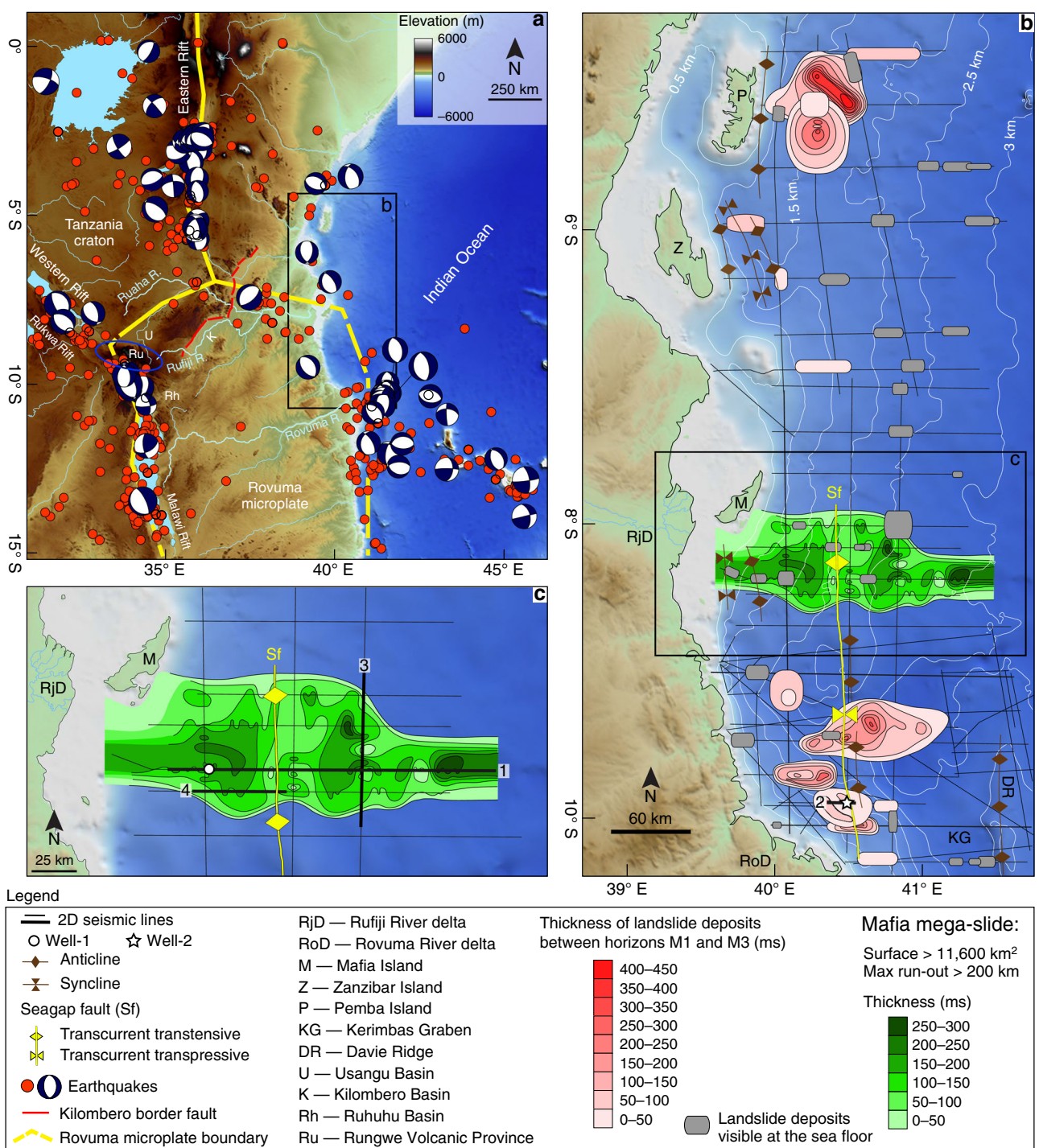

**Fig. 1 Maps of the study area with main structural elements, available seismic and well data, and distribution of submarine landslides. a** Digital Elevation Model of East Africa and western Indian Ocean (data from the General Bathymetric Chart of the Oceans—GEBCO and ref. [49]) with distribution of earthquakes with magnitude ≥4.0 for the time period 1976–2018 (red circles) and Global Centroid Moment Tensor solutions[65]. The blue ellipse outlines the Rungwe Volcanic Province. **b** Bathymetry of the western Indian Ocean offshore Tanzania with highlighted landslide deposits, major structural elements, and the seismic dataset (black lines). **c** Close-up view of the Mafia mega-slide (50-ms-spaced isopach). The locations of Well-1 and Well-2 are marked by a white circle and a star, respectively. The thick black lines show the locations of the seismic profiles 1–4, presented in Figs. 3, 4, 5, and 6.

which mainly corresponds to an erosional surface offshore Mafia Island, as highlighted by reflection truncations (Figs. 3, 5). To the south, offshore the Rovuma delta, M2 is mainly conformable (Fig. 4) and embedded in a shale-rich sedimentary succession, as highlighted by low-amplitude reflections and high Gamma-Ray (GR) values (Fig. 2). Horizon M2 is encountered at 2129 m ssTVD in Well-1, ca. 140 meters below the base of the Mafia

mega-slide, and at a depth of 3171 m ssTVD in Well-2 (Fig. 2). Horizons M3 and M4 each correspond to a continuous positive reflection, laterally changing in amplitude and becoming conformable towards deeper waters. M4 is at places eroded by active deep-water channels, which are also visible at the seabed, or by submarine landslides (Figs. 3, 5). Horizon M3 ties with Well-1 at 1655 m ssTVD and with Well-2 at 2380 m ssTVD, while M4 ties

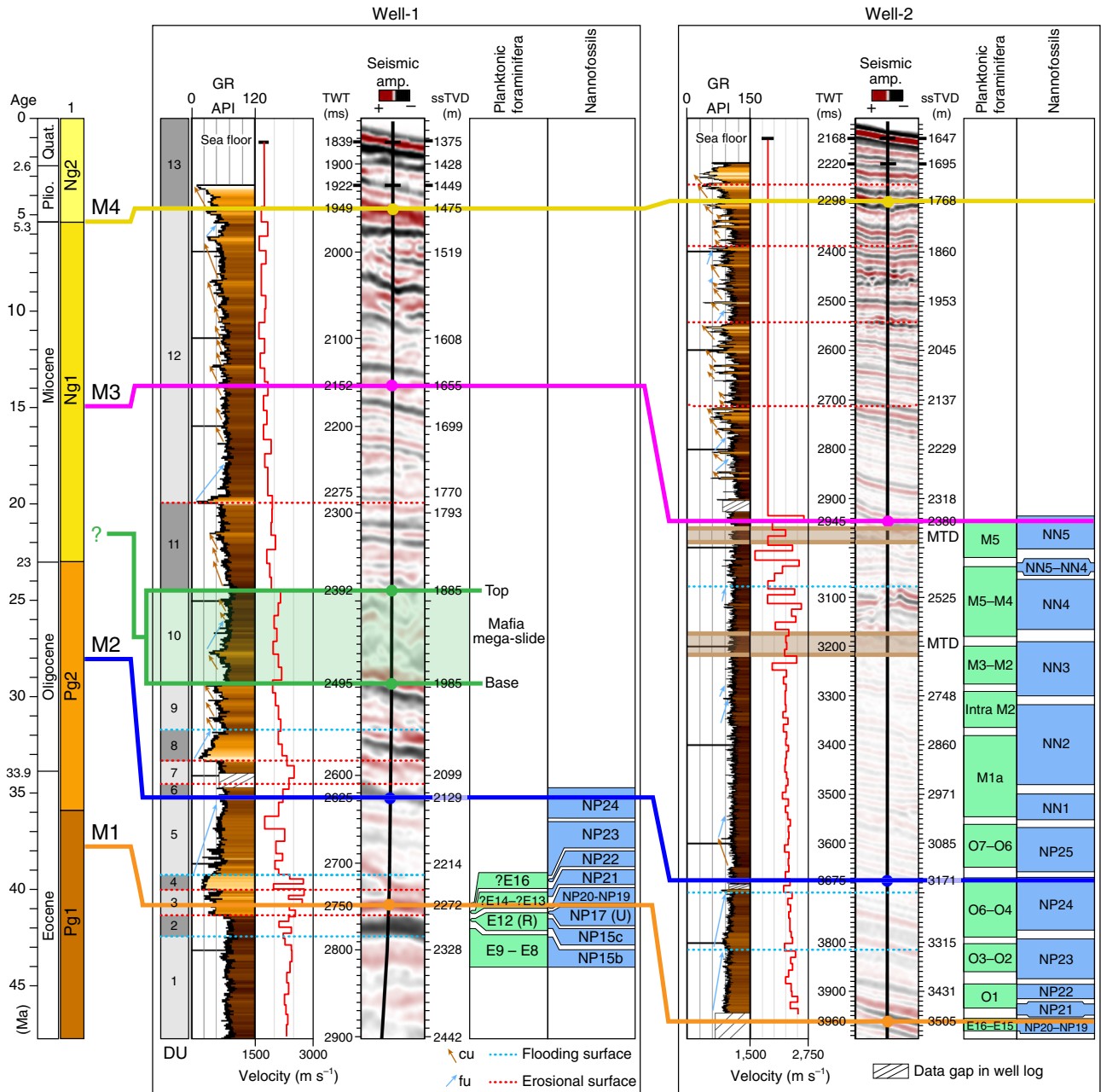

**Fig. 2 Well-to-seismic tie and biostratigraphy of Well-1 and Well-2.** Well-to-seismic tie of Well-1 and Well-2 with dated stratigraphic horizons (M1–M4), seismic amplitude at the well sites, Gamma-Ray (GR), velocity model from check-shots, depth below sea level in two-way travel time (TWT, milliseconds) and true vertical depth (ssTVD, meters), planktonic foraminifera and nannofossil zones. The Mafia mega-slide is highlighted in green. 1: Stratigraphic sequences from ref. [42]; DU: depositional units (Supplementary Note 2); fu: fining-upward, cu: coarsening-upward, MTD: mass-transport deposit.

at depths of 1475 m and 1768 m ssTVD with Well-1 and Well-2, respectively (Fig. 2).

The chronology of horizons M1–M4 (Fig. 2) has been derived through biostratigraphic data from the exploration wells (Supplementary Tables 1–4) and correlation with dated horizons presented in previous studies[13,14,39,43–46]. Horizons M1 and M2 encounter both wells where biostratigraphic data are available; M1 dates back to the Priabonian (upper Eocene), as suggested by nannofossil evidence indicative of Zone NP20-NP19[47], while M2 dates to the Rupelian–Chattian (Oligocene) Zone NP24[47] (Fig. 2, Supplementary Tables 2 and 4). Horizon M3 is dated in Well-2, where nannofossils are indicative of mid-Miocene Zone NN5[47] (Supplementary Table 4). Horizon M4 is not biostratigraphically constrained in the wells, but it correlates with horizon H3

of ref. [14] that was dated to the early Pliocene by using foraminifera assemblages. M4 marks the base of sequence Ng2 defined in ref. [42].

**Mafia mega-slide**. In the offshore Rufiji River delta (Fig. 1), the most notable deposit in the post-Eocene stratigraphy is an up to 300-milliseconds (ms)-thick seismic unit (corresponding to ca. 300 m considering an averaged interval velocity of 2000 m s⁻¹), mainly characterized by a chaotic to transparent seismic reflection configuration (Figs. 2, 3, 5, 6, Supplementary Note 2, Supplementary Figs. 2 and 3). The basal surface shows an erosive character (Figs. 3, 5), as highlighted by its irregular morphology discordant with respect to the underlying stratigraphy and by step-like features cutting the sediment beneath (Fig. 3b). The top,

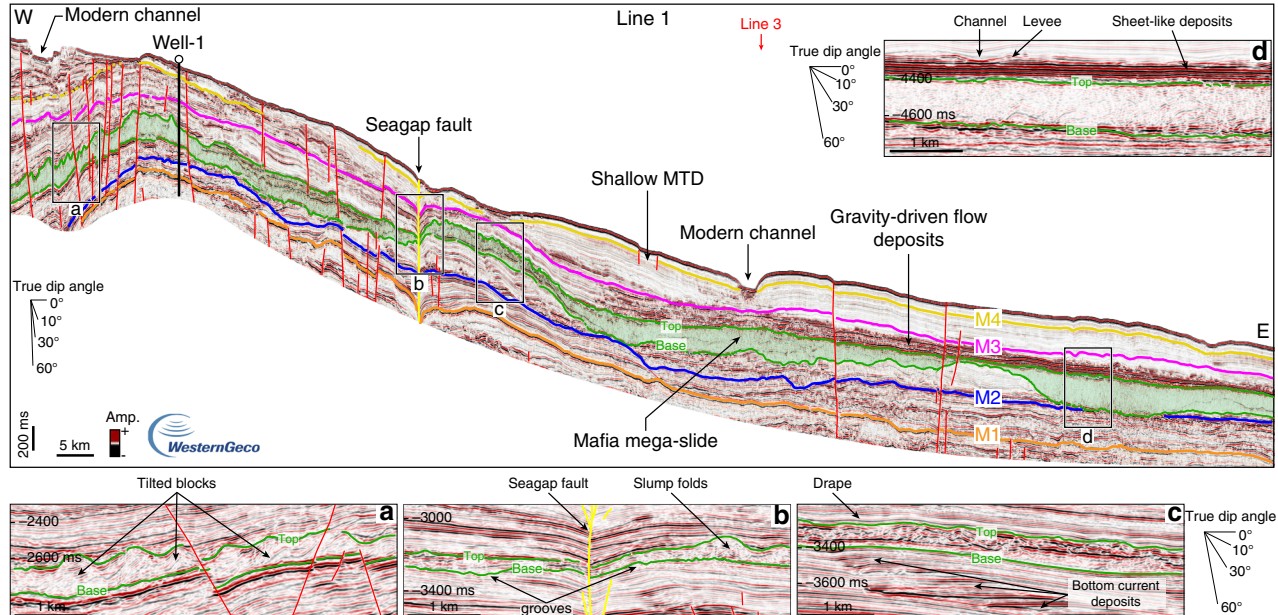

**Fig. 3 Stratigraphy of the Tanzania margin imaged by a downslope-oriented seismic line crossing Well-1.** Seismic Line 1, oriented W-E and intersecting seismic Line 3 (red arrow), is perpendicular to the slope of the western Indian Ocean offshore the Rufiji River delta (location in Fig. 1). Insets **a–d** highlight the morphological and seismic expression variability of the Mafia mega-slide and bounding units. M1–M4 are the main seismic horizons identified in this study and tied to the two wells (see Fig. 2 for the ages). The Mafia mega-slide is highlighted in green. MTD: mass-transport deposit. Noninterpreted version of Line 1 is available in the Supplementary Fig. 2.

marked by a positive (red) reflection, also shows a significant topographic relief (Fig. 3 and Supplementary Fig. 3). The unit is visible for the entire length of the seismic profile, ca. 200 km, in the downslope direction (W-E, Fig. 3), and it extends along slope (N–S) over a distance of up to ca. 80 km (Figs. 1c, 5). Locally, this seismic unit shows steeply dipping reflections (Fig. 3a), high-amplitude hummocky to contorted reflections (Fig. 3b, c), and low-amplitude discontinuous to chaotic reflections (Fig. 3d, Supplementary Fig. 3). The basal surface intersects Well-1 at a depth of 1985 m ssTVD, while the top is found at depth of 1885 m ssTVD (Fig. 2). The GR values show a sudden increase across the basal surface, consistent with its erosive nature (Fig. 2), and indicate that the 100 m thick sequence is characterized by fine-grained sediments, with the highest mud content (high GR) across the upper-surface. Intervals with low GR may indicate the presence of thin sandstones or reworked shallow-marine carbonates, the latter probably belonging to Paleocene–Eocene sequences[43]. From its internal stratal reflection configuration, its external geometry, the erosive nature of the basal surface, and the overall stratigraphic context we interpret this unit as a mass-transport deposit (MTD) resulting from a giant submarine landslide or mass wasting event. The shaley nature of the deposit (high GR values) and its thickness variation also indicate that the landslide most likely originated by the collapse of the upper and fine-grained slope south-east of the modern Mafia Island (Fig. 1c).

Submarine landslides originate when the applied shear stress exceeds the shear strength of slope-forming materials, and the mass of sediments starts moving downslope under the action of gravity[2]. The onset of movement is often favoured by the presence of weak layers[7] and can be triggered by different mechanisms[5,6,8]. On seismic data, MTDs are characterized by a variety of seismic facies, ranging from chaotic or highly disrupted internal reflection configurations to packages of coherent internal reflections[3]. The different seismic facies that characterize the Mafia mega-slide, such as high-amplitude hummocky reflections, contorted reflections, and steeply dipping coherent internal

reflections, can be related to the presence of slump folds and tilted blocks within the deposit[48]. The step-like features along its basal surface correspond to erosional scours, grooves, or glide tracks generated during its emplacement[48]. The Mafia mega-slide covers an area of more than 11,600 km$^2$ and has a volume of at least 2500 km$^3$ (Fig. 1c). The seismic profiles available do not allow us to map the full extent of the mega-slide, nor the head scarp that is probably located below the modern Rufiji River delta or even further inland. The toe also is not visible in the dataset. Consequently, the run-out distance of the mega-slide is greater than 200 km, and its toe lies in the abyssal plain, in modern water depth >3 km.

Well-to-seismic tie and correlation with dated seismic horizons presented in previous studies allow us to date the emplacement of the Mafia mega-slide (Fig. 2): it is bounded by late Oligocene deposits at its base (horizon M2) and by Burdigalian to Langhian (early to middle Miocene) deposits above (horizon M3).

**Other submarine landslide deposits.** Seismic data reveal the presence of several other units on the Tanzania margin characterized by chaotic seismic facies that are associated to large landslide deposits (Fig. 1). These MTDs, for which the areal extent and thickness have been determined (Fig. 1b), are visible in the southern part of the Tanzanian margin, offshore the Rovuma River delta (Fig. 4), and towards the north, offshore Pemba Island (Fig. 1). In southern Tanzania, the age of these deposits is constrained by Well-2, and the oldest deposits date back to the Burdigalian (early Miocene), as also discussed in ref. [43]. The age determination of the MTDs in northern Tanzania in contrast is less straightforward due to the lack of accessible well data in the area. However, lateral correlation of horizons M1 and M2 suggests that the MTDs offshore Pemba Island (Fig. 1b) can be as old the Eocene. Smaller submarine landslides (with run-out distances <30–50 km) are widespread across the margin, also on the modern sea floor (Fig. 1b), and have been identified by combining seismic profiles with bathymetric data[49]. In the

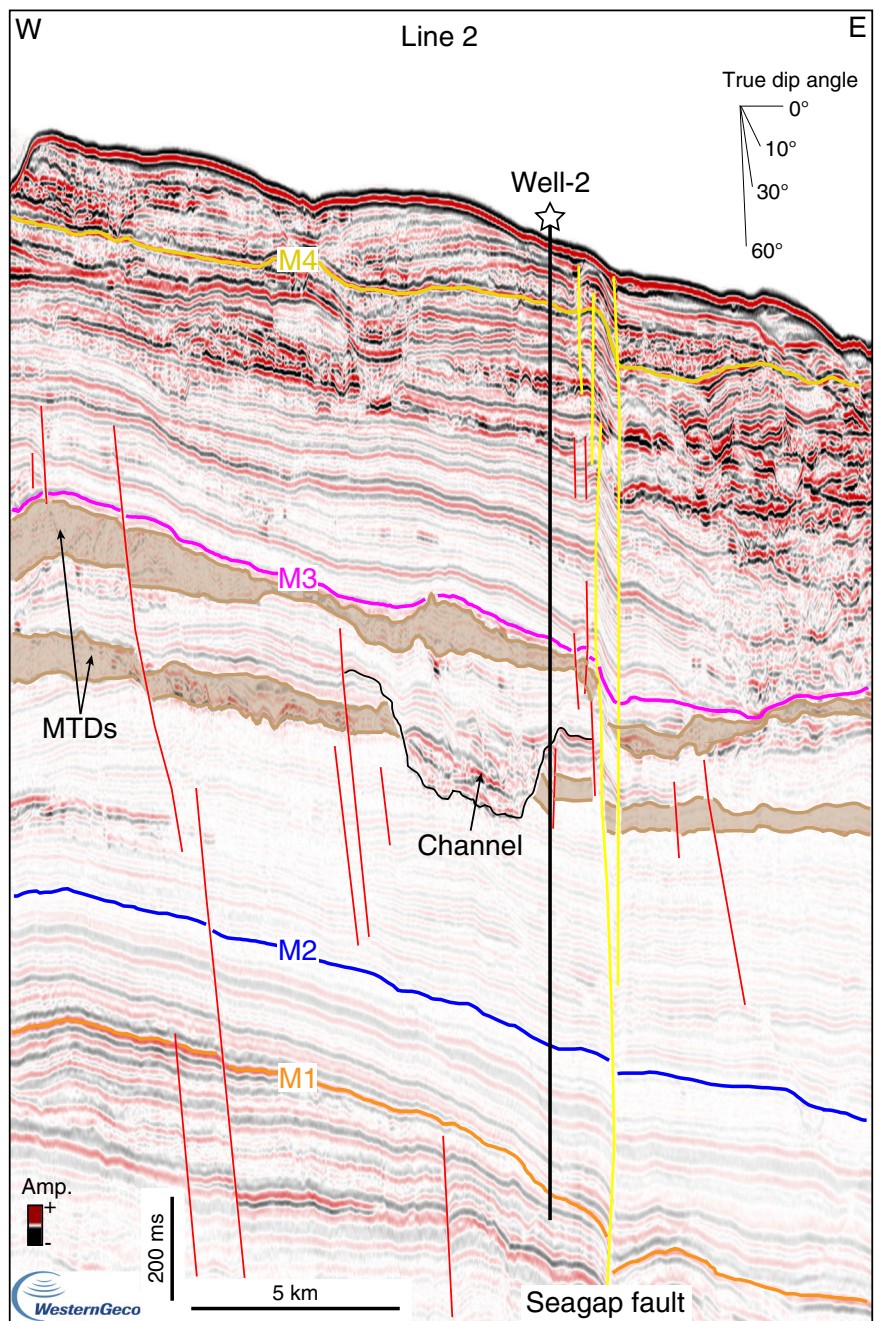

**Fig. 4 Stratigraphy of the Tanzania margin imaged by a downslope-oriented seismic line crossing Well-2.** Seismic Line 2, oriented W-E, crosses Well-2 offshore the northern Rovuma River delta. In this area, the oldest major mass-transport deposits (MTDs) accumulated during the Burdigalian (see Fig. 2 for the ages of the horizons). Noninterpreted version of Line 2 is available in the Supplementary Fig. 5.

available data, no large (run-out distances greater than 100 km) landslides are visible in the stratigraphy above horizon M4, in Pliocene to recent deposits. Detailed age characterization of the age of all the landslides is beyond the scope of this work, and will require future investigations.

## Discussion

Considering the age and location of the Mafia mega-slide a question arises: Were the Mafia mega-slide and the other large mass-transport deposits linked to the tectonic activity of the East African Rift System? The plateau and rift-flank uplift of the EARS radically changed the topography of East Africa, reorganizing river catchments, drainage basins, and moderating climate[23,31]. Microfaunal and pollen assemblages sampled in exploration wells

from the Rovuma delta offshore Mozambique indicate a change towards a cooler and wetter climate through the middle Eocene to Oligocene[50], during the crustal uplift related to the EARS[41]. Recent studies from the western Indian Ocean also highlight the EARS' influence on offshore sedimentation along the margins of Tanzania and Mozambique[13,14]. The evolution of broad flexural uplifts on the rift shoulders promoted the formation of endorheic basins and reorganized pre-existing basins, such as the reactivated Rukwa and Malawi basins, redirecting the flow of the Ruaha–Rufiji and Rovuma Rivers to the east[23,33,51,52]. This is consistent with the paleo-flow directions, directed to E-ESE, observed in the Mikindani Formation, interpreted as the post-Oligocene paleo-Rovuma deltaic complex[40]. Relatively rapid volcanic construction in the Rungwe Volcanic Province since

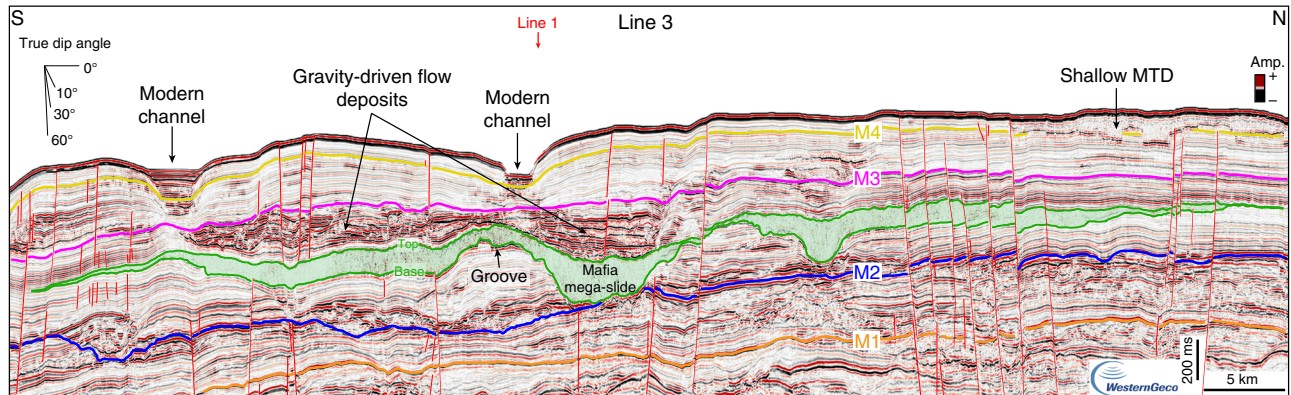

**Fig. 5 Seismic line showing the along-strike variability of the Mafia mega-slide and bounding deposits.** Seismic Line 3, oriented N–S and intersecting seismic Line 1 (red arrow), shows continuous to discontinuous and high-amplitude reflections between the top of the Mafia mega-slide and horizon M3, that are interpreted as coarse-grained gravity-driven flow deposits. MTD: mass-transport deposit. Noninterpreted version of Line 3 is available in the Supplementary Fig. 6.

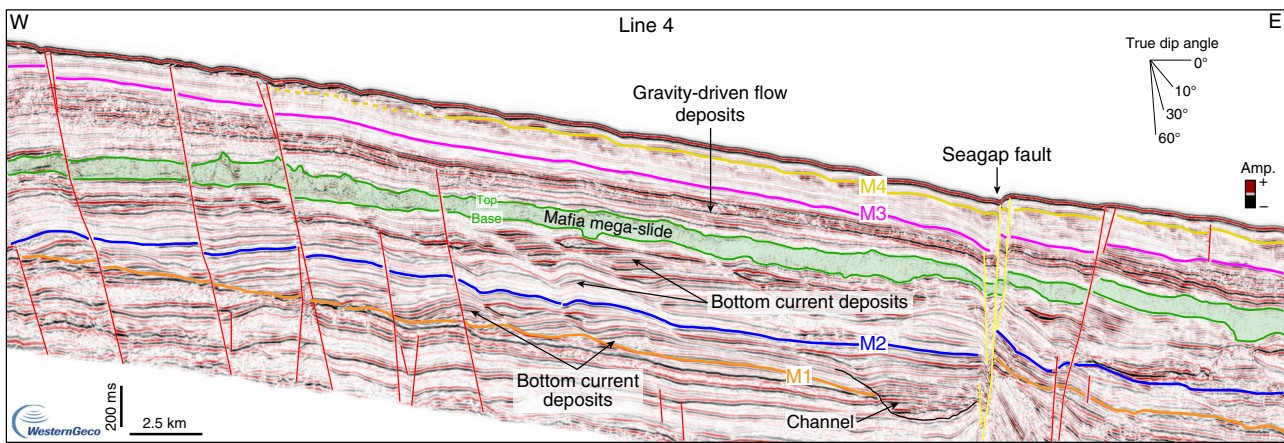

**Fig. 6 Downslope-oriented seismic line across the Mafia mega-slide.** Seismic Line 4, oriented W-E, shows: (1) high-amplitude and wavy reflections with an upslope direction of progradation (to the west) between horizon M1 and the base of the Mafia mega-slide, that are interpreted as bottom current deposits (i.e., sediment waves formed by ocean circulation); (2) continuous to discontinuous and high-amplitude reflections, mainly sub-parallel, between the top of the Mafia mega-slide and horizon M3, that are interpreted as coarse-grained gravity-driven flow deposits; (3) continuous and low-amplitude reflections between M3 and M4, that are interpreted as shaley deep-water pelagic deposits and that become more influenced by ocean bottom currents above M4, where sediment waves are again visible. Noninterpreted version of Line 4 is available in the Supplementary Fig. 7.

19 Ma also diverted the drainage[24]. Sediment supply towards the Indian Ocean had been increasing since the Eocene[41] due first to prerift doming and then rift-flank uplift and volcanic topographic changes, leading to the progradation of the Rufiji and Rovuma River deltas and the over-steepening of the slope, as suggested by the architecture of the turbidite channels[43]. A change in provenance in the epidote-dominated sandstones of the Mandawa Basin has been documented between the middle Eocene and the Oligocene[53], suggesting that uplift and increased fluvial discharge affected the continental drainage patterns and catchment areas of the Palaeo–Rufiji and Paleo-Rovuma Rivers. If strain localization between the Tanzania craton and the Rovuma microplate initiated at 25–20 Ma with initial Western Rift magmatism, the Ruaha–Rufiji drainage basin would have sourced reworked sediments from the Usangu, Kilombero, and Ruhuhu Karroo basins[23,33].

Earthquake and fault patterns provide additional constraints on the cause of the Mafia mega-slide. Observations and theory indicate that earthquake hazard is greatest in strong, thick lithosphere in extension, where large stresses can be stored. Moment magnitude is proportional to the fault area, which will be largest in cold lithosphere with greater seismogenic thickness,

as in the Western Rift where seismicity spans the ~40 km thick crust[12]. For example, one of the largest instrumentally recorded earthquakes to affect Africa was the 2006 Mw 7.0 Machaze earthquake in Mozambique. The earthquake occurred in thick and largely unfractured lithosphere in a new zone of rifting at the edge of the Zimbabwe craton[54], causing surface ruptures along an approximate 50-km-long zone and up to 1 m surface displacement[9,10,54]. The thick cratonic lithosphere beneath central Africa would have been colder and stronger at rift initiation, indicating that Mw > 7.0 earthquakes likely characterized Oligo-Miocene rift development onshore. The passage of seismic waves from East African rift earthquakes may have triggered submarine sediment failures, based on analogy to dynamic triggering of landslides in the Gulf of Mexico[5]. Regional earthquakes in oceanic lithosphere beneath the Kerimbas Graben and Davie Ridge may also have triggered landslides[9,13,55,56]. Reference [36] noted a ca. 12 km wide headwall of a massive landslide along the Miocene Kilombero border fault (Fig. 1), indicating that large-scale landslides occurred along seismically active faults approximately at the same time. The combination of the higher elevation of the rift flanks and seismically active, steep, long, extensional fault systems in weakly indurated Karroo strata (and in some areas, Paleogene strata, and

tropical weather systems) likely led to temporary damming and catastrophic release of sediment-clogged river systems, boosting sediment supply to the western Indian Ocean. Rapid sediment accumulation, over-steepening of the sea floor, and earthquakes all potentially contributed to the generation of the Mafia mega-slide, and probably of the other large landslide deposits of similar age observed offshore the Tanzania. This conclusion is aligned with the results of other studies on submarine landslides in other continental margin settings[57]. The presence of only smaller-scale MTDs in more recent times is consistent with the reduction in sedimentation rate observed for Plio-Quaternary deposits in the western Indian Ocean[41] (Supplementary Note 3 and Supplementary Fig. 4). The timing of the Mafia mega-slide is broadly constrained between ca. 28 Ma and 15 Ma by the age of horizons M2 and M3, respectively, but a more precise age can be quantified considering the stratigraphic relationship between the Mafia mega-slide and horizon M3. Because the top of the landslide is 230 meters below M3 and the sedimentation rate in the interval 28–15 Ma is between ~29 m Myr$^{-1}$ and 48 m Myr$^{-1}$ (Supplementary Note 3 and Supplementary Fig. 4), the landslide likely occurred between 22.9 Ma and 19.8 Ma. The emplacement of large MTDs in southern Tanzania only during the Burdigalian (late early Miocene) is in agreement with an early-mid Miocene emplacement of the Mafia mega-slide (Fig. 2). A similar timing for MTDs offshore Mafia Island has also been suggested[43]. If the above interpretations are correct, the emplacement of large MTDs coeval with initial volcanism in the Western Rift and dynamic uplift of the East African margin in the western Indian Ocean lends further support to initial faulting, magmatism, and Mw > 7.0 earthquakes in some sectors of the EARS by early Miocene times[24,26,32]. The Rukwa–Malawi Rift was the site of extension in Permo–Triassic time, and its location between two cratons may have enhanced plume-lithosphere interactions in this region[58,59].

The Mafia mega-slide represents the largest single depositional event in the post-Eocene history of the Tanzanian margin, and most likely generated a tsunami wave that flooded many low-lying coastal zones facing the Indian Ocean. In comparison with other submarine landslides worldwide[60], the Mafia mega-slide is one of the biggest deposits discovered so far (Fig. 7). Its size is comparable with the Storegga landslide, a submarine landslide that originated on the Norwegian slope causing a tsunami that left sand deposits on many coastal sites along the western side of the North Sea[61].

The bathymetric changes due to the emplacement of the Mafia mega-slide influenced the evolution of the deep-water sediment routing system and the deep-ocean circulation for millions of years, as demonstrated by the drastic change in the seismic reflection configuration before and after the deposit (Figs. 3c, 6). In detail, while the pre-slide stratigraphy shows a series of lenticular high-amplitude reflections often associated with more wavy and low-amplitude reflections (Fig. 3c), the post-slide stratigraphy shows high-amplitude reflections onlapping onto the topography generated by the slide deposit in a proximal position, where Well-1 is located (Fig. 3, and Supplementary Fig. 3), and an up to 150 ms thick unit showing high lateral continuity and high-amplitude reflections farther downslope (Fig. 3d) or southward (Fig. 6). The pre-slide deposits are interpreted as deep-water turbidite lobes and sediment waves formed by the interaction of gravity-driven flows and ocean bottom currents[43,62] (Figs. 3c, 6, Supplementary Notes 1 and 2), whereas the post-slide units have properties indicative of reduced bottom current activity and a dominance of gravity-driven deposition (Figs. 3d, 5, Supplementary Notes 1 and 2). This is also highlighted in the well data by repeated coarsening-upward units in the Miocene section (Fig. 2). The physiographic change of the slope resulting from the Mafia mega-slide promoted the formation of new turbidite

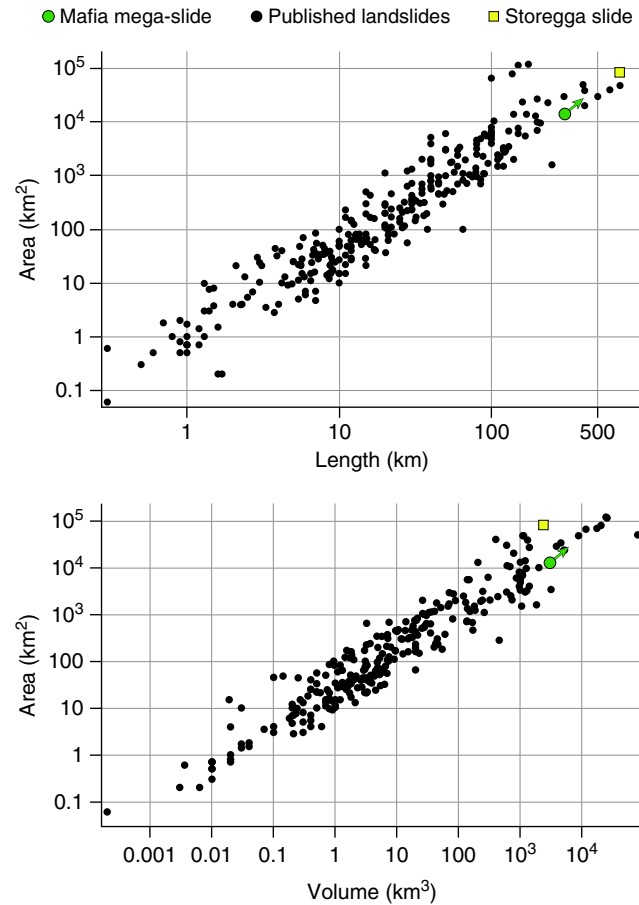

**Fig. 7 Morphometry of submarine landslides globally.** Landslide area/length and area/volume ratios for published landslide deposits in comparison with the Mafia mega-slide. Data from ref. [60].

channels, as highlighted by the presence of high-amplitude and high-continuity reflections, normally associated with sheet-like turbidite deposits and channel-levee systems (Fig. 3d). A reduced bottom current activity inferred from the seismic reflection configuration of the post-Oligocene deposits is also noticed outside the area affected by the Mafia mega-slide: in offshore Tanzania, ref. [43] interpreted the cessation of the southward migration of the channel systems at the start of the Miocene as due to reduced, or even absent, bottom currents, while offshore Kenya and Somalia, ref. [63] identified deep-water deposits influenced by bottom currents primarily in upper Cretaceous and Paleogene sequences. This evidence indicates that other processes may have influenced the western Indian Ocean circulation at the same time. Lateral thickness changes of the Mafia mega-slide (Fig. 1c) not only may reflect post-depositional erosional processes but also the control of the paleo-topography generated by the tectonic activity at the time of deposition.

Our results demonstrate that large and potentially tsunamigenic landslide events are associated with plateau uplift and continental rifting in East Africa. In addition, active faulting visible at the sea floor[14] points to the possibility that submarine landslides, likely of a reduced extent, may still occur, representing an important hazard to this area of fast coastal population growth and large development of offshore infrastructures. Indeed, the recognition of several episodes of sediment failures in recent times requires additional investigations to quantify the recurrence interval of such events, and the potential trigger mechanisms. Our data emphasizes the need for future

research aimed to evaluate the risk associated with submarine landslides in the region.

## Methods

**Well data.** Gamma-Ray (GR) log and biostratigraphy information from two exploration wells were made available for this study by Tanzania Petroleum Development Corporation (TPDC), Royal Dutch Shell, and Shell Tanzania, and were used to characterize the lithology and chronology of the area (Figs. 2, 3, 4). Well-1 was drilled in 1375 m water depth offshore the Rufiji River delta, the GR starts about 74 m below the sea floor at a true vertical depth below mean sea level (ssTVD) of 1449 m, while micro and nannopaleontology data are only available below 2000 m ssTVD, for sediments older than the upper Oligocene (Supplementary Fig. 3). Well-2 was drilled in 1647 m water depth offshore the northern Rovuma River delta, the GR starts about 48 m below the sea floor at 1695 m ssTVD, while micro and nannopaleontology data are available below 2300 m ssTVD, for sediments older than the middle Miocene (Supplementary Fig. 5). Biostratigraphic data from the two wells are presented in Supplementary Tables 1–4. Sonic or density data are not available at the two wells to construct synthetic seismograms, but a detailed check-shot surveys enables time-depth calibration between the wells and seismic data (Supplementary Note 2).

In Well-1, we identified 13 depositional units (DU) over the ~1000 m thick section of interest and four key seismic horizons (M1–M4 from deep to shallow), which were mapped throughout the study area and correlated with Well-2. The units are described in detail in the Supplementary Note 2 and Supplementary Fig. 3.

**Seismic data.** Two seismic surveys were made available for this study: the multi-client Tanzania 2D seismic dataset, acquired by WesternGeco-Schlumberger in 1999–2000, and the GLOW seismic dataset (Paleogene GLObal Warming events, GLOW Cruise[64]), performed onboard of the R/V Pelagia in 2009. The seismic lines presented in this study are from the multi-client Tanzania 2D survey (Figs. 3–6), which consists of 11,000 km of seismic lines (5550 km available for this study) acquired by using a 5200 m long streamer length with hydrophones at a 12.5 m receiver interval. The data were reprocessed in 2012 applying an anisotropic Kirchhoff prestack time migration that improved signal resolution, fault definitions, and events continuity, thus providing higher confidence when interpreting stratigraphic features. The vertical resolution of the seismic data in the investigated section ranges between 7.5 m and 12 m, calculated considering a peak frequency of 60 Hz and interval velocities of 1800–2900 m s$^{-1}$. The seismic data are zero-phase, with the water bottom reflection, associated with a positive reflection coefficient, expressed as a strong red half-cycle between two weaker, black, sidelobes (Fig. 2). The GLOW seismic survey[64] consists of 2450 km of seismic lines, acquired by using an array of four airgun sources (10, 20, and 2 × 40 in$^3$) and a 24-channel streamer as a receiver, consisting of four 63 m long active sections with six channels each. The seismic data were recorded using the GeoResources Geo-Trace 24 system. The far field signal of the source array shows a peak frequency centred within the range of 50–150 Hz, with a frequency content up to 400 Hz. The vertical resolution of the seismic data in the investigated section ranges between 2.5 and 5 m, calculated considering a peak frequency of 150–200 Hz and interval velocities of 1800–2900 m s$^{-1}$.

Noninterpreted versions of the seismic profiles presented in Figs. 3–6 are provided in Supplementary Figs 2, 5, 6, and 7.

## Data availability

The authors confirm that all relevant data are included in the paper and its Supplementary Information file. Original (noninterpreted) seismic profiles and the source data for the chronology of Well-1 and Well-2 are provided in the Supplementary Figs. 2, 5, 6, and 7, and Supplementary Tables 1–4. The bathymetric data acquired during the GLOW cruise are available in the open access library PANGAEA. All additional data are available upon reasonable request to the authors. Morphometric data of submarine landslides are available online at the S4SLIDE project website (https://sites.google.com/a/utexas.edu/s4slide/home), ref. [60].

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

## Acknowledgements

We are grateful to the Tanzania Petroleum Development Corporation (TPDC), WesternGeco and Schlumberger, Royal Dutch Shell, and Shell Tanzania for giving access to the seismic and well data and allowing the publication of this work. We would also like to thank Schlumberger for providing academic licenses of the seismic interpretation software Petrel, and J. van Wijk for sharing complementary work in review. The Paleogene GLObal Warming events (GLOW) cruise, onboard of RV Pelagia, was funded by the ESF EUROCORES program (NWO project number 855.01.122). V.M. was supported by the Ocean Frontier Institute, through an award from the Canada First Research Excellence Fund, and by the NSERC Discovery Grant. C.J.E. work was supported by NSF grant 1734884 (Malawi Rift project with new compilations). B.S.W. was supported by the Joint Oceanographic Institutions/US Science Support Program and NERC grant NE/G014817. P.N.P. was supported by NERC-UK-IODP grant NE/F523293/1.

## Author contributions

V.M., corresponding author, devised the conceptual idea, wrote much of the paper, and outlined the figures. C.J.E. helped in defining the relation between the Mafia mega-slide and the tectonics of the East African Rift System. D.I., C.J.E., and D.K. contributed to the writing of the manuscript. D.I., S.T., H.d.H., M.F., A.v.V., and B.R. supported seismic data interpretation. B.S.W. and P.N.P. helped in the interpretation of biostratigraphic data. All the authors contributed to the final editing and revision of the manuscript.

## Competing interests

The authors declare no competing interests.
