## [Peer Review File · Nature Communications]

Reviewers' comments:

Reviewer #1 (Remarks to the Author):

The Authors present data on the stratigraphic succession of the passive margin offshore Tanzania that suggest the presence of huge submarine landslide (named Mafia mega-slide), which likely occurred during the Late Oligocene or Early Miocene. The Authors suggest that this mega-slide could have been triggered by tectonic activity related to the East African Rift, specifically by large earthquakes and enhanced sediment supply. The topic is interesting and of broad relevance, therefore suited for publication in Nature Communications. However, I think there are some aspects that have to be improved significantly before the paper can be considered further.

#1. The main problem I see is related to the discussion and interpretation of results, which I think should be significantly improved in terms of relations with tectonic processes. Indeed, I think there is a lack of detailed comparison between the timing of the mega-landslide and that of volcanic-tectonic activity in the basin and/or plateau uplift, which is the main focus of the discussion section (see question in lines 212-214). In this respect the text is very general, with no specific indications of the ages of faulting and volcanic activity in the basins west of the landslide region (see below point #2) – there is only a rough indication of ages in lines 232-234 or 256 but I think more data could be taken from published works (also given that one of the Authors -Ebinger- has a worked in the area for a long time). It is also not very clear if the mega event could have been linked to plateau uplift in East Africa rather than tectonic activity in the single basins. So, the discussion of these aspects should be significantly improved.

#1b. This discussion could be extended, if possible, to other submarine landslides recognized in the area. Is their occurrence related to periods of intense tectonic activity recognised in the nearby basins?

#2. Related to this, the geological setting is rather poor and could be improved. For instance, the Authors should say something more about the timing of plateau uplift in East Africa, providing some details about the debate on its timing. Also, the Authors should present a detailed analysis (from literature data) of the timing of tectonic and volcanic activity in the area of interest (e.g., Rukwa basin, Rukwa volcanic province).

#3. I suggest to move the Methods section at the end of the paper (as I have seen in many Nat Comms papers).

Minor points

Line 70. I would remove the term 'Main' here

Line 73. Occurrence of an offshore branch of the EARS (Franke et al) could be maybe mentioned here.

Line 73 and throughout the manuscript. Many geographical terms reported in the main text are not indicated in Figures. For instance: Okavango; Usangu; Ruhuhu...

Lines 128-132. This is not clear to me. To be improved.

Line 259. Maybe a reference to the work by Roberts et al. 2012 or some review papers (e.g., Macgregor 2015 J Afr Earth Sciences) could be appropriated here

Reviewer #2 (Remarks to the Author):

This manuscript reports on the discovery of a very large submarine landslide off the East African coast, named the Mafia megaslide. The landslide is recognized in seismic reflection data, that clearly show its chaotic, hummocky character. The landslide is probably upper Oligocene in age.

The cause of the landslide is related by the authors to onset of rifting in East Africa, which created topography that increased the sediment supply, which, combined with earthquake activity, may have triggered the landslide. The landslide is thought to have changed the ocean circulation in the western Indian Ocean for the remainder of the Neogene.

This topic is of interest to a broader audience and therefore suitable for publication in a journal with a broad reader base such as Nature Communications. I have a few questions that the authors can easily address. A more important point is the public availability of the data presented here. I did not find a statement on data availability in the files provided to me for review, so I refer this concern back to the editor.

1) The cause of this landslide (earthquake activity and large sediment supply) is speculative, but plausible. The authors may want to refer to recent work by Fan, McGuire, and Shearer, who have a manuscript in press at GRL on triggered landslides in the Gulf of Mexico. Their work very nicely shows that quite large submarine landslides can be triggered by mid-size earthquakes, quite far from the earthquake location. This has, I think, implications for the discussion on the cause of the landslide in this manuscript- it may not be related to the onset of long normal faults in the young rift zone (which is in my opinion a weak point of the manuscript- I would expect immature fault zones in the young rift zone, which would not create large earthquakes). Mid-size earthquakes on shorter faults, also further away from the landslide location, may have triggered the event.

2) The age of the landslide is not well constrained, nor is the source location. Could Foraminifera or nannofossils from the landslide itself provide information on displaced sediments and their original depositional location, and provide a more precise age constraint on the landslide? Are these data available? If so, how do they constrain the landslide dynamics?

3) "The mega-slide changed the basin geometry of the western Indian Ocean, altering the deep-water sediment routing system and affecting ocean circulation for the remainder of the Neogene."

Sufficient evidence is not provided in the manuscript to support this claim. In lines 267-285, local changes are described, so I would limit this discussion to the location of the landslide. The changes described are likely mostly caused by the changed bathymetry; I do not see evidence in the manuscript for a change in western Indian Ocean circulation. Inset d in fig 3 shows as far as I can see, a reflection pattern at the top of the box which is similar to the pre-landslide reflection pattern toward the left of the box (about 5 km mid-box). If no further evidence is provided for a change in western Indian Ocean currents, I suggest to remove the reference to a change in western Indian Ocean circulation here and in the abstract.

l. 169 south-east : should this be south-west?

Reviewer #3 (Remarks to the Author):

This is a clear, well-written and concise paper which does a valuable job of bringing the large-scale Mafia Slide to the attention of a wider audience. The lay-out flows well and the diagrams are particularly good, clear and well-annotated, and clearly support the observations made in the text. I have uncertainty regarding the age of the Mafia Slide as being as old as Late Oligocene, but this is based on the larger 3D seismic and well datasets that I have had access to in the past. I have attached a regional geoseismic image that was presented in my 2017 PESGB/HGS poster and partly in my 2018 paper, which puts the Mafia Slide in the context of horizon age interpretations that have been projected northwards from well control in Blocks 1 and 2. The base of the Ng1 (Base Miocene) sequence is well constrained up to the northern edge of Block 2. North of here, the shallowest biostratigraphic control is from the Early Oligocene. I have chosen to take the Base Miocene beneath the Mafia Slide, because the southern tip of the slide (in the northern part of Block 3) is quite high up in the stratigraphy, and I would not interpret the Base Miocene as being this shallow. This line also shows the two other large slides in Blocks 1 and 2, which appear to be at a similar stratigraphic level. The oldest major slide in Block 1 is post Burdigalian, and I have suggested a Middle Miocene age. I would question why supplementary figure S2 is not included in the main paper, as it beautifully illustrates the rotated slide blocks in the vicinity of the well. I have suggested a couple more references that you might want to review with respect to the offshore EARS system in Tanzania and the slope channel systems: Mulibo and Nyblade 2019 and Fongnesu et al. 2020 (details in the manuscript pdf with comments).

Best regards, Pamela Sansom, 1/3/2020

Halifax, 27 April 2020

This letter presents a point-by-point reply to all the comments we received from three reviewers. For clarity, the comments are in red and our replies in black.

- Reply to Reviewer 1 -

1- The main problem I see is related to the discussion and interpretation of results, which I think should be significantly improved in terms of relations with tectonic processes. Indeed, I think there is a lack of detailed comparison between the timing of the mega-landslide and that of volcanic-tectonic activity in the basin and/or plateau uplift, which is the main focus of the discussion section (see question in lines 212-214). In this respect the text is very general, with no specific indications of the ages of faulting and volcanic activity in the basins west of the landslide region (see below point #2) – there is only a rough indication of ages in lines 232-234 or 256 but I think more data could be taken from published works (also given that one of the Authors -Ebinger- has worked in the area for a long time). It is also not very clear if the mega event could have been linked to plateau uplift in East Africa rather than tectonic activity in the single basins.

1- Following the comment of the reviewer, we have modified both the interpretation and the discussion as follows. In the results, we provide two additional seismic lines, one offshore the Rufiji delta and one offshore the northern Rovuma delta. The Rufiji seismic line shows the post-Eocene stratigraphy and in particular the marked change in the depositional style, from a dominance of sediment-waves and bottom current deposits below the landslide to deposition mainly driven by turbidity flows above it. Offshore the northern Rovuma delta, the new seismic line which also crosses the second exploration well allows to date the stratigraphy up to the mid-Miocene. The seismic line shows that submarine landslide deposits are only present in Miocene sediments, thus better constraining the age of the mass failure events across the margin. In the ‘Geological Setting’ section, we present a better overview of the available constraints on the age of rifting and topographic uplift in East Africa. In the discussion, we use this new information to support our model that margin instability was linked to the EARS. We also included a description of the evolution of the Rungwe volcanic province (headwater of the Rufiji river basin) and of the Mikindani Formation, interpreted as the post-Oligocene paleo Rovuma deltaic complex (lines 245-255). The comment of why *the mega event could have been linked to plateau uplift rather than tectonic activity in the single basins*, is not fully clear to us. Is the reviewer referring to rift basins onshore? As demonstrated in many other studies, a combination of processes may contribute to the generation of submarine landslides, and those can be grouped in two families: preconditioning factors and trigger mechanisms (see Masson et al., 2006). The onset of the EARS in Tanzania and

associated plateau uplift (first) and rift-flanks uplift (later) have changed the drainage network of the rivers redirecting the flow, and sediment supply, towards the Indian Ocean. As said above, now we provide a better overview of published data on this (lines 85-140). Over-steepening of the margin due to the tectonics of the EARS and increased sediment supply acted as preconditioning factors, while the strong earthquakes that are associated to the early stages of rifting were the potential trigger. This is, for us, the best explanation for the occurrence of large landslide deposits in the Miocene. The new seismic line offshore the northern Rovuma shows that, in the proximity of the well location, there are no landslide deposits in the interval between ca. 38 Ma (even earlier, as this is the age of horizon M1, dated to the Priabonian) and ca. 20 Ma.

2- This discussion could be extended, if possible, to other submarine landslides recognized in the area. Is their occurrence related to periods of intense tectonic activity recognised in the nearby basins?

2- Our study is the first presenting a detailed chronology of submarine landslides in the post-Eocene stratigraphy of the Tanzania margin. Sansom (2018) also noticed a potential link between landslide and rifting, and we have acknowledged her work. No studies are available for the margins of Kenya and Somalia, while the few studies which are mainly focusing in the offshore Mozambique only investigate Late Quaternary sediments.

3- Related to this, the geological setting is rather poor and could be improved. For instance, the Authors should say something more about the timing of plateau uplift in East Africa, providing some details about the debate on its timing. Also, the Authors should present a detailed analysis (from literature data) of the timing of tectonic and volcanic activity in the area of interest (e.g., Rukwa basin, Rukwa volcanic province).

3- We agree with this comment and have expanded the 'Geological Setting' by including a critical review of the literature available on the timing of plateau uplift (lines 81-120). We also provide additional data on the onshore stratigraphy of the Paleo-Rovuma deltaic deposits to support our model. As suggested by the reviewer, we have also included the chronological data available on the formation of the Rukwa volcanic province, where the headwaters of the Rufiji river are located (lines 107-120). We have modified Figure 1 to include also this additional information.

4- I suggest to move the Methods section at the end of the paper (as I have seen in many Nat Comms papers).

4- Yes, the Methods section now appears after the Discussion (lines 350-394).

- Reply to Reviewer 2 -

1- The cause of this landslide (earthquake activity and large sediment supply) is speculative, but plausible. The authors may want to refer to recent work by Fan, McGuire, and Shearer, who have a

manuscript in press at GRL on triggered landslides in the Gulf of Mexico. Their work very nicely shows that quite large submarine landslides can be triggered by mid-size earthquakes, quite far from the earthquake location. This has, I think, implications for the discussion on the cause of the landslide in this manuscript- it may not be related to the onset of long normal faults in the young rift zone (which is in my opinion a weak point of the manuscript- I would expect immature fault zones in the young rift zone, which would not create large earthquakes). Mid-size earthquakes on shorter faults, also further away from the landslide location, may have triggered the event.

1- We were not able to find the paper by Fan et al. on GRL mentioned by the reviewer. We contacted Prof Shearer and he confirmed that the paper is still in review, so it cannot be used as reference in this work. Said that, we partially agree with this comment. Despite we cannot fully rule out that mid-size earthquakes away from the landslide location might have played a role, observations and theory indicate that earthquakes' magnitude is greatest in strong thick lithosphere in extension, where large stresses can be stored. For example, one of the largest instrumentally recorded earthquakes to affect Africa was the 2006 Mw 7 Mozambique EQ that occurred in largely unfractured rock in a new zone of rifting characterized by relatively thick cold lithosphere at the edge of the Zimbabwe craton (Yang and Chen, 2008). As a consequence, large earthquakes are the most likely trigger of the offshore landslides seen in Oligo-Miocene deposits. We have modified the manuscripts to clarify this point (lines 291-300).

2- The age of the landslide is not well constrained, nor is the source location. Could Foraminifera or nannofossils from the landslide itself provide information on displaced sediments and their original depositional location, and provide a more precise age constraint on the landslide? Are these data available? If so, how do they constrain the landslide dynamics?

2- We have no age data from the landslide itself but we present solid age information from below the mega-slide and the ages of the main reflectors M2 and M3 are known (see Fig. 2), resulting in an age of early-mid Miocene by extrapolation using the seismic lines. We further constrain the ages above and below landslide activity on this margin by referring to another carefully dated well in the south and the mean sedimentation rates of the slope region for the interval 28-15 Ma. In the first submission, we presented data from one exploration well where biostratigraphic data are available for Eocene and Oligocene deposits below the mega-slide. As a consequence, there are no foraminifera or nannofossils age data from the well above this specific landslide deposit to provide information on the age of displaced sediments and their original depositional location. In this new manuscript, however, we present a second exploration well located farther to the south, offshore the northern Rovuma Delta, where also Miocene sediments are recovered, and thus detailed biostratigraphic (chronological) data are available up to the Langhian (see new Figure 2 and supplementary information). Seismic and well data (new Figures 2 and 4) show that landslide deposits in this area of well 2 occur during the lower to mid Miocene interval, thus providing new additional constraints on when the collapse of the margin started (lines 280-288). This conclusion also agrees with the results presented by Sansom (2018).

3- “The mega-slide changed the basin geometry of the western Indian Ocean, altering the deep-water sediment routing system and affecting ocean circulation for the remainder of the Neogene.” Sufficient evidence is not provided in the manuscript to support this claim. In lines 267-285, local changes are described, so I would limit this discussion to the location of the landslide. The changes described are likely mostly caused by the changed bathymetry; I do not see evidence in the manuscript for a change in western Indian Ocean circulation. Inset d in fig 3 shows as far as I can see, a reflection pattern at the top of the box which is similar to the pre-landslide reflection pattern toward the left of the box (about 5 km mid-box). If no further evidence is provided for a change in western Indian Ocean currents, I suggest to remove the reference to a change in western Indian Ocean circulation here and in the abstract.

3- We agree with the comment of the reviewer, we meant to say that the bathymetric changes caused by the Mafia mega-slide affected the local ocean bottom circulation. We have reworded this sentence in the text and in the abstract (lines 37-40 and lines 322-333). In addition, it is important to note that there is a broader change in ocean bottom currents moving from the Oligocene to the Miocene that can be seen all along the margin of the western Indian Ocean, from Tanzania to Somalia. This change is noticed by Coffin and Rabinowitz (1988) on well and seismic data and also by Sansom (2018). Many processes may have contributed to this change, including the closure of the Tethys and the formation of Antarctica Ice Sheet with implications for bottom water currents, but are not discussed in detail in the present study.

- Reply to Reviewer 3, Pamela Sansom -

1- I have uncertainty regarding the age of the Mafia Slide as being as old as Late Oligocene, but this is based on the larger 3D seismic and well datasets that I have had access to in the past. I have attached a regional geoseismic image that was presented in my 2017 PESGB/HGS poster and partly in my 2018 paper, which puts the Mafia Slide in the context of horizon age interpretations that have been projected northwards from well control in Blocks 1 and 2. The base of the Ng1 (Base Miocene) sequence is well constrained up to the northern edge of Block 2. North of here, the shallowest biostratigraphic control is from the Early Oligocene. I have chosen to take the Base Miocene beneath the Mafia Slide, because the southern tip of the slide (in the northern part of Block 3) is quite high up in the stratigraphy, and I would not interpret the Base Miocene as being this shallow. This line also shows the two other large slides in Blocks 1 and 2, which appear to be at a similar stratigraphic level. The oldest major slide in Block 1 is post Burdigalian, and I have suggested a Middle Miocene age.

1- After the first submission of this paper, we secured access to additional data from Block 1 which are the ones cited by the Reviewer. These new data, which are presented in the new version of the manuscript and Supplementary Information, gave the possibility to improve the chronology of the post-Eocene succession of the western Indian Ocean and to better constrain the Mafia mega-slide. We were able to correlate a new seismic horizon, which is dated in Block 1, up to the area where the landslide is located. The timing of the Mafia mega-slide is now constrained between ca. 28 and 15 Ma if we only use age of horizons M2 and M3, and between 19.8 and 22.9 Ma if we consider the sedimentation rate (lines 280-288 and Supplementary Information). We agree with the Reviewer

that the oldest major slide in Block 1 is post Burdigalian, and that the Mafia mega-slide likely occurred during the early-mid Miocene. We have modified the text accordingly. The Reviewer provided a series of comments and edits in a separate PDF file, and we have included all of them in the new version of the manuscript.

We are extremely grateful to the reviewers for all their suggestions, which greatly improved the quality of the manuscript.

Yours sincerely,

Vittorio Maselli

Dalhousie University

Department of Earth and Environmental Sciences

REVIEWERS' COMMENTS:

Reviewer #1 (Remarks to the Author):

The Authors have improved the manuscript that is to me almost ready for publication.

A minor point:

I do not clearly understand the description of earthquake characteristics in lines 299-308. This treatment is to me not strictly necessary and is –at least in parts- a repetition of what already introduced in lines 261 and following. So, I suggest to either remove lines 299-308 (or merge them with the description in lines 261 and following). Anyway, I do not see the need to describe some details of earthquakes (such as the dip of the fault plane of the 2006 Machaze event)

Reviewer #2 (Remarks to the Author):

I read the new version of the manuscript and the response of the authors to the questions of the reviewers. I believe that the authors did an excellent job addressing all concerns.

I found two typos in the manuscript:

- l. 144: change manly to mainly
- l. 386: change metres to meters

Reviewer #3 (Remarks to the Author):

The re-submitted paper is considerably improved by the additional well data and seismic from Well-2. This addresses and supports the comments I made regarding dating of the Mafia slide in the first review and also provides more information on the presence and generally similar age of other major slides in the area. It has also been improved by the additions to address the comments of the other two reviewers. I would like to re-iterate the very high quality and clarity of the figures in this manuscript.

Please see attached versions of the main text and supplementary material where I have made some minor text-edit comments, including some strange characters that I think have come through instead of hyphens. I also think that you have the seismic lines 1 and 3 labelled the wrong way round in Fig. 1c.

Pamela Sansom

Halifax, 12 June 2020

This letter presents a point-by-point reply to all the comments we received from three reviewers. For clarity, the comments are in red and our replies in black.

- Reply to Reviewer 1 -

I do not clearly understand the description of earthquake characteristics in lines 299-308. This treatment is to me not strictly necessary and is –at least in parts- a repetition of what already introduced in lines 261 and following. So, I suggest to either remove lines 299-308 (or merge them with the description in lines 261 and following). Anyway, I do not see the need to describe some details of earthquakes (such as the dip of the fault plane of the 2006 Machaze event)

We agree with the comment made the reviewer and we have modified lines 299-308 and merged them with the paragraph starting with line 261, avoiding to duplicate information. We have also removed the details of the Machaze event.

- Reply to Reviewer 2 -

I found two typos in the manuscript: l. 144: change manly to mainly and l. 386: change metres to meters

Thank for noticing the typos. We have fixed them in the new version.

- Reply to Reviewer 3, Pamela Sansom -

The re-submitted paper is considerably improved by the additional well data and seismic from Well-2. This addresses and supports the comments I made regarding dating of the Mafia slide in the first review and also provides more information on the presence and generally similar age of other major slides in the area. It has also been improved by the additions to address the comments of the other two reviewers. I would like to re-iterate the very high quality and clarity of the figures in this manuscript. Please see attached versions of the main text and supplementary material where I have made some minor text-edit comments, including some strange characters that I think have come through instead of hyphens. I also think that you have the seismic lines 1 and 3 labelled the wrong way round in Fig. 1c.

We appreciate the kind comment made by the reviewer on the quality of the figures. Thank you. The new version of the manuscript also includes all the comments provided by the reviewer in the

PDF files of the manuscript and supplementary information. We have also modified Figure 1, thank you for noticing this mistake on the numbering of the seismic lines.

We are grateful to the reviewers for the additional effort made in reviewing our manuscript.

Yours sincerely,

Vittorio Maselli

Dalhousie University

Department of Earth and Environmental Sciences